# Enhanced Anomaly Detection System for IoT Based on Improved Dynamic SBPSO

**DOI:** 10.3390/s22134926

**Published:** 2022-06-29

**Authors:** Asima Sarwar, Abdullah M. Alnajim, Safdar Nawaz Khan Marwat, Salman Ahmed, Saleh Alyahya, Waseem Ullah Khan

**Affiliations:** 1Department of Computer Systems Engineering, University of Engineering and Technology, Peshawar 25120, Pakistan; asima.sarwar@uetpeshawar.edu.pk (A.S.); safdar@uetpeshawar.edu.pk (S.N.K.M.); sahmed@uetpeshawar.edu.pk (S.A.); waseem@uetpeshawar.edu.pk (W.U.K.); 2Department of Information Technology, College of Computer, Qassim University, Buraydah 51452, Saudi Arabia; 3Department of Electrical Engineering, College of Engineering and Information Technology, Onaizah Colleges, Onaizah 56447, Saudi Arabia; saleh.alyahya@oc.edu.sa

**Keywords:** anomaly detection, Internet of Things, intrusion detection system, IoT security

## Abstract

The Internet of Things (IoT) supports human endeavors by creating smart environments. Although the IoT has enabled many human comforts and enhanced business opportunities, it has also opened the door to intruders or attackers who can exploit the technology, either through attacks or by eluding it. Hence, security and privacy are the key concerns for IoT networks. To date, numerous intrusion detection systems (IDS) have been designed for IoT networks, using various optimization techniques. However, with the increase in data dimensionality, the search space has expanded dramatically, thereby posing significant challenges to optimization methods, including particle swarm optimization (PSO). In light of these challenges, this paper proposes a method called improved dynamic sticky binary particle swarm optimization (IDSBPSO) for feature selection, introducing a dynamic search space reduction strategy and a number of dynamic parameters to enhance the searchability of sticky binary particle swarm optimization (SBPSO). Through this approach, an IDS was designed to detect malicious data traffic in IoT networks. The proposed model was evaluated using two IoT network datasets: IoTID20 and UNSW-NB15. It was observed that in most cases, IDSBPSO obtained either higher or similar accuracy even with less number of features. Moreover, IDSBPSO substantially reduced computational cost and prediction time, compared with conventional PSO-based feature selection methods.

## 1. Introduction

With the rise of the Internet, there has been an immense surge in Internet-based services [1]. As a result, many of the physical systems or devices that are connected to the Internet can easily be operated and managed remotely. Client behaviour can then be monitored and documented, future decisions can be predicted, and useful services provided [2]. The Internet of Things (IoT) is used in a variety of fields, including the smart home, smart city, smart healthcare, smart factories, smart supply chain, and smart retail. Figure 1 depicts a few of IoT applications that may be found in everyday life. The goal of such a smart environment is to make people’s lives more productive and add value by addressing issues related to living conditions [3]. However, because of increased interconnectedness, the network has become more complicated, making network security more difficult to sustain. Intruders consider security lapses to be an invitation to discover and exploit vulnerabilities in IoT networks. However, network security breaches can result in significant financial losses for businesses and consumers. Hence, it is essential to design a system that will ensure the security of the IoT network. Many tools and techniques are available to combat various cyber-attacks, such as spam filters, firewalls, anti-malware, intrusion detection systems (IDSs), intrusion prevention systems (IPSs), and so on [4].

To ensure the security of an IoT network, an IDS can be an extremely effective and crucial solution. There are three key phases in the operation of an IDS. The first of these is monitoring, which is based on network or host sensors. The second phase is analysis, which involves feature extraction and pattern recognition. Finally, the third phase is detection, which detects any anomalies in a network.

Intrusion detection systems can be classified into two main groups: signature-based intrusion detection systems (SIDS) and anomaly-based intrusion detection systems (AIDS). Traditional SIDS methods involve examining network packets and attempting to match patterns to a signature database. A machine-learning (ML) approach is used in AIDS to train the model in normalised behaviour. Network activities are then compared with that normal behaviour. Anomaly-based intrusion detection systems are considered as a dynamic approach to anomaly detection, applying behaviour-oriented detection.

The AIDS strategy has in fact received more attention than any other approach [5]. The capacity to detect unknown or zero-day attacks is the main benefit of AIDS. The majority of researchers choose anomaly detection, since it appears to be the most viable means [6,7]. However, designing efficient IDS for IoT devices remains challenging, due to the following reasons:(a)**Cyber-security datasets**The majority of existing datasets are outdated and may be inefficient for grasping the behavioural patterns of modern cyber-attacks. Moreover, there is a dearth of knowledge about the characteristics of recent attacks and their patterns of occurrence.(b)**Handling quality problems in Cyber-security datasets**Cyber-security datasets may be incomplete, unbalanced, noisy, or contain inconsistent instances related to a particular security incident. The quality of the learning process, and performance of ML-based models is affected by such dataset issues [8].(c)**Low processing ability**Internet of Things devices are lightweight and energy-constrained with low computational capacity. However, teal-time data-processing is required by ML algorithms, which presents a problem to the implementation of such resource-constrained devices.(d)**Low memory capacity**Data is created in diverse ways in the IoT context, necessitating huge memory in IoT devices. As a result, being able to offer an efficient solution for varied data poses a hurdle.

Moreover, employing all features in the design of an IDS can lead to the introduction of redundant and irrelevant features into the model. Therefore, feature optimization must be used to achieve good IDS performance [9]. There are three main approaches to feature optimization. The filter-based approach evaluates features according to predefined metrics, often using information theory. In contrast, a wrapper and embedded approach will evaluate features using an ML algorithm. In this current study, a wrapper-based feature optimization technique was used, specifically IDSBPSO as it gives efficient results as compare to other feature optimization methods [10]. SBPSO is a recently proposed BPSO variant that updates a particle’s position, using the flipping probability rather than velocity. In SBPSO, a stickiness parameter is employed to maintain the momentum that is characteristic of PSO, meaning that a particle will tend to adhere to the position to which it has recently moved. PSO is a population-based stochastic optimization algorithm. Due to its easy feature-coding, computational reasonability, few parameters, and less demanding execution to address and select critical feature problems, the PSO algorithm is considered efficient and robustness to control parameters. There are various publicly available datasets for IoT networks, which include DARPA98, KDDCUP99, CAIDA (2007), ISCX 2012, ADFA-WD (2014), ADFA-LD (2014), CISIDS 2017, DS2OS (updated 20218), BOT-IoT (updated 2020) UNSW-NB15, and IoTID20.

The following are the contributions of the paper:The proposed IDSBPSO is based on a novel approach of dynamic bit-masking strategy to reduce the search space of the SBPSO. This approach iteratively applies a mask to features after a certain number of generations, in order to prevent those features from evolving further. Using such a method throughout the evolutionary process can significantly reduce the search space, allowing the IDSBPSO to identify better solutions within a smaller search space.Some parameters are set to dynamic, in order to investigate how this strategy can help balance exploration with exploitation, thereby further improving the searchability of SBPSO for the problems of optimising feature selection.The proposed strategy would be implemented on two IoT network datasets for feature optimization, since this strategy is proposed for the design of an anomaly detection system for IoT networks, as a means of reducing the computational cost of such networks when using devices of a constrained nature.

The proposed FS model will be tested on the 2 datasets, IoTID20 and UNSW-NB15. The proposed model obtain comparable or higher accuracy with reduced computational cost and less number of features compared to benchmark PSO based methods. The remainder of this paper is organised as follows: the literature review is presented in Section 2; Section 3 discusses the proposed framework architecture; Section 4 describes the implementation and evaluation of the results of the system experiments, and Section 5 concludes the paper, also making recommendations for future work.

## 2. Literature Review

Internet of Things network security remains a consistent research topic for security analysers. Hence, numerous IDSs have been proposed, based on various types of feature optimization and reduction methodologies. In [11], the authors propose a novel two-tier classification model based on ML methodologies, for example, the Naive Bayes, K nearest neighbors (KNN) classifier with certainty factor voting, and linear discriminant analysis (LDA) for feature reduction. This model has a high detection rate for sophisticated attacks like User to Root (U2R) and Remote to Local (R2L), namely 34.81% and 67.16%, respectively. Conversely, in [12], the authors propose an effective deep learning approach: a self-taught learning (STL) IDS. The NSL-KDD dataset was used in the above-mentioned study, but the authors suggest a hybrid method for more accurate results. In [13], the authors suggest a feature selection technique using filter and wrapper methods, but these are computationally expensive. Meanwhile, in [14], the authors propose three IDS on K-means clustering, a decision tree, and a hybrid of these methods to achieve a maximum detection rate of 70–93%.

In [15], however, the authors propose a hybrid deep network, combining Convolutional Neural Network (CNN) with a gated recursive unit to detect intrusion. A PSO algorithm was utilised in the resulting study to select relevant features from the data, and a developing system successfully performed the feature selection and classification process automatically. Meanwhile, in [16], the authors present a semi-supervised ML technique for distributed denial of service (DDoS) detection, based on network entropy estimation, co-clustering, information gain ratio, and an extra-tree algorithm. This demonstrated good accuracy but with increased complexity. Conversely, in [17], the authors employed a variety of feature selection strategies, including a correlation coefficient, gain ratio, and information gain. The suggested experiment was carried out on random forest, rotation forest, and random committee classifiers.

Meanwhile, in [18], the authors present a feature selection-based IDS. The feature classification algorithm was based on a linear correlation coefficient. The cuttlefish algorithm was also used in this method to select features based on filter and wrapper, respectively. The FCC-CFA (feature grouping according to the linear correlation coefficient-cuttlefish algorithm) approach was created to extract the optimal subset of features from the dataset. This is a hybrid form of filter and wrapper method, retaining the advantages of each. The KDD Cup99 dataset was then used to test the suggested approach. The results of utilising the FGLCC-CFA algorithm revealed that compared with the CFA and FGLCC algorithms, the hybrid method was able to improve the accuracy and detection rate, while also reducing the number of false alarms.

In contrast, using a two-phase approach, the authors in [19] propose a hybrid intrusion detection model. Here, the first phase consisted of feature selection and the second, detecting an attack. A wrapper method called MGA-SVM was applied in the first phase. With multi-parent crossover and multi-parent mutation, this model combines the characteristics of SVM and GA (MGA). In the second phase, an artificial neural network (ANN) was used to detect attacks, and PSO was employed to improve the suggested model’s performance. The proposed name of this model is therefore MGA-SVMHGS-PSO-ANN. It has a high detection accuracy of 99.3%, according to data from the NSL-KDD dataset.

On the other hand, specifically for lightweight IoT devices, the performance of a lightweight ML-based IDS was tested in [20], using a new feature selection technique. The technique was verified with a public dataset, acquired from an IoT environment for this work. In the above model, a new feature selection approach, referred to as correlated-set thresholding on gain-ratio (CST-GR), is proposed to create a lightweight system, while also positively affecting the detection rate.

In [21], however, the authors propose supervised ML algorithms to create a three-layer intrusion detection system, capable of detecting a variety of cyber-attacks in IoT networks. The resulting solution was tested in a smart home scenario with eight IoT gadgets. In [22], the authors designed a bottom-up EI architecture and proposed novel data driven dynamical control strategy. Moreover, Intelligent controllers augmented by deep reinforcement learning (DRL) techniques are adopted and the concept of curriculum learning (CL) is integrated into DRL to improve the sample efficiency and accelerate the training process. Similarly, in [23], the authors created a novel hybrid intrusion detection system (HIDS) for IoT threats. The developed HIDS ensemble was utilised to secure IoT devices by merging SIDS with AIDS. The results of the generated model revealed that the HIDS was superior in its performance. Conversely, the model could not detect various types of attack on the IoT system.

According to the research cited above, various FS methods have been used in the past but when the data dimensionality increases then it cause serious challenge for optimisers, as search space increases dramatically. Choosing the right characteristics to maximise classification accuracy for anomaly detection in IoT networks, while at the same time reducing computational cost and prediction time, would still appear to be a work in progress. Various research exists on the design of anomaly detection systems for IoT networks, but these either use benchmark PSO-based methods, or a hybrid of optimization algorithms for feature selection. To close the gap in the literature, this study therefore provides an intelligent system, which uses novel approach to reduce search space and increase the exploration and exploitation ability of optimizer to select optimal features, while obtaining comparable or higher accuracy with reduced computational cost and prediction time.

## 3. The Proposed Model

This section proposes an enhanced approach to the design of an efficient and accurate IDS for IoT networks, using an IDSBPSO as an approach to feature selection. Particle swarm optimization (PSO) is a population-based stochastic optimization algorithm, proposed by Eberhart and Kennedy in 1995 [24]. Because of its easy feature-coding, computational reasonability, few parameters, and less demanding execution to address and select critical feature problems, the PSO algorithm is considered efficient [25]. The originally proposed PSO was a continuous one (CPSO), used to tackle a variety of continuous issues. The main drawback of PSO is that if a particle gets stuck in a local minimum (optimal), all the other particles will converge to that local minimum, resulting in erroneous solutions. Thus, before expanding the network, it is necessary to preserve particle diversity [26].

Particles are employed in the PSO method to represent solutions from the population of particles in the relevant space. This population is referred to as a swarm. Each particle in the swarm is represented by vector $x_{i}=(x_{i}{,}_{1},x_{i}{,}_{2},$…$,x_{i}{,}_{d})$, where *d* represents the number of features in the dataset, and each particle has *d* dimensional velocity $v_{i}=(v_{i}{,}_{1},v_{i}{,}_{2},$…$,v_{i}{,}_{d})$. To enhance efficiency, PSO works randomly and travels in the search space to find relevant features by updating velocity and position with iterations. At each iteration, the particles’ velocity and position are updated according to pbest and gbest, which are the best personal and global fitness values up until that iteration. According to [27], the position and velocity of particles is updated as in (1) and (2).
(1)$$v_{i,d}^{k+1}=wv_{i,d}^{k}+c_{1}r_{1}\left(pbest_{i,d}^{k}-x_{i,d}^{k}\right)+c_{2}r_{2}\left(gbest_{i,d}^{k}-x_{i,d}^{k}\right)$$
(2)$$x_{i,d}^{k+1}=x_{i,d}^{k}+v_{i,d}^{k+1}$$
where *k* represents $k_{th}$ iteration and *d* represents $d_{th}$ feature in the vector space. In addition, *w* represents the inertia factor that will give weightage to the previous velocity, and $c_{1}$ and $c_{2}$ are acceleration coefficients that give weightage to the cognitive and social term in the updated velocity. Meanwhile, $r_{1}$ and $r_{2}$ are uniform random numbers within [0, 1].

Velocity has three components, as illustrated in (1). The first component is momentum, depicting the influence of the present direction. Varying particles usually have different momentums, which help keep the swarm diverse, especially when everyone shares their finest experiences. Furthermore, momentum is the only factor that will allow a particle to continue seeking better solutions, once it has arrived at the best point discovered by the swarm so far. Conversely, the other two are cognitive and social components which guide particles towards an optimal experience, as well as that of each particle’s neighbours.

Binary PSO was developed to solve combinatorial problems, including job-shop scheduling and feature selection. In BPSO, rather than adding velocity to position, in order to obtain a new position, velocity is used to determine the probability of achieving the corresponding updated position values [27], which can be seen in (3).
(3)$$x_{d}^{k+1}=\left\{\begin{array}{cc} 1 & rand\left(\right)\leq s\left(v_{i,d}^{k+1}\right)\\ 0 & otherwise \end{array}\right.$$
(4)$$s\left(v_{d}^{k+1}\right)=1/e^{-v_{d}^{k+1}}$$

Sticky BPSO (SBPSO) is a recently proposed BPSO variant that updates a particle’s position, using the flipping probability rather than velocity. In SBPSO, a stickiness parameter is employed to maintain the momentum that is characteristic of PSO, meaning that a particle will tend to adhere to the position to which it has recently moved [28]. This is illustrated in (5).
(5)$$x_{d}^{k+1}=\left\{\begin{array}{cc} 1-x_{i,d}^{k} & rand\left(\right)\leq p_{i,d}^{k+1}\\ x_{i,d}^{k} & otherwise \end{array}\right.$$
where rand () is a random value in [0, 1] from the uniform distribution. Moreover, $p_{i,d}^{k+1}$ is the flipping probability of the $i^{th}$ particle in the *d* dimension [28], which may be written mathematically as per (6).
(6)$$p_{i,d}^{k+1}=n_{s}\left(1-st_{i,d}^{k}\right)+n_{p}\times |pbest_{i,d}-x_{i,d}^{k}|+n_{g}\times \left|gbest_{d}-x_{i,d}^{k}\right|$$
where $st_{i,d}^{k}$ denotes the stickiness parameter of the $i^{th}$ particle on the $d^{th}$ dimension. Here, $pbest_{i,d}$ denotes the personal best of the $i^{th}$ particle on the $d^{th}$ dimension, and $gbest_{d}$ denotes the global best. Meanwhile, $n_{s}$, $n_{p}$, and $n_{g}$ are the three parameters that give weightage to the particle’s stickiness ability and its tendency to move towards pbest and gbest. The stickiness parameter, $st_{i,d}^{k}$ lowers over time, indicating that a bit is more likely to cling to its new position. According to [28], the updated $st_{i,d}^{k}$ mechanism is illustrated in (7).
(7)$$st_{i,d}^{k}=\left\{\begin{array}{cc} s_{i,d}^{k}-1⁄M & x_{i,d}^{k+1}=x_{i,d}^{k},s\left(v_{i,d}^{k}\right)>0\\ 1 & x_{i,d}^{k+1}\neq x_{i,d}^{k} \end{array}\right.$$
where *M* is the step parameter determining stickiness ability, which decreases from 1 to 0 as the number of iterations increases. Initially, $s_{i,d}^{k}=1$ was set for *k* = 0. Dynamic SBPSO is a further variant of the SBPSO variant, proposed to control the exploration and exploitation ability of particles. In dynamic SBPSO, $n_{s}$, $n_{p}$, and $n_{g}$ are used to increase exploration at the outset and increase exploitation at the end. Here, $n_{s}$ and $n_{g}$ linearly decrease in relation to an increase in the number of iterations, which can be seen in (8) and (10), respectively. Meanwhile, $n_{p}$ linearly increases alongside the rising number of iterations, which can be seen in (9).
(8)$$n_{s}=n_{s_{max}}-k/k_{max}\times \left(n_{s_{max}}-n_{s_{min}}\right)$$
(9)$$n_{p}=n_{p_{min}}+k/k_{max}\times \left(n_{p_{max}}-n_{p_{min}}\right)$$
(10)$$n_{g}=n_{g_{max}}-k/k_{max}\times \left(n_{g_{max}}-n_{g_{min}}\right)$$
where $n_{s_{max}}$ and $n_{s_{min}}$ are the maximum and minimum values for the $n_{s}$ factor, $n_{p_{max}}$ and $n_{p_{min}}$ are the maximum and minimum values for the $n_{p}$ factor, and $n_{g_{max}}$ and $n_{g_{min}}$ are the maximum and minimum values for the ng factor. Ultimately, *k* represents the $k^{th}$ iteration, and kmax represents the maximum number of iterations. The values applied for all these parameters can be seen in the subsection, ‘Parameter Setup’.

Traditionally, during the evolutionary process, a BPSO algorithm searches in a fixed *d*-dimensional space (where *d* represents the number of original features). When *d* is large, setting a high number of particles or generations in the PSO algorithms demands significant processing resources. As a result, it is advantageous to include a search space reduction strategy, which can lower the computational resources required for the PSO applied to the feature selection task.

In this study, the dynamic bit-masking strategy was combined with DSBPSO. This first involved extracting information from the pbests of particles to determine which bits should be masked. During the evolutionary process, the number of selected traits of all particles decreases. Even before the halting criterion is met, noisy or irrelevant features can be determined. After a certain number of generations, if a feature (bit) is not selected by all pbests in the swarm, it is very probable that this feature is useless, since solutions containing this feature are very likely to be eliminated for their poor fitness. The parameter that decides when a mask should be updated is μ. In this study, the mask update approach was adopted, because a bit is masked if it is not selected by all pbests in the swarm. This can be seen in Algorithm 1.
**Algorithm 1:** Search Space Reduction Strategy
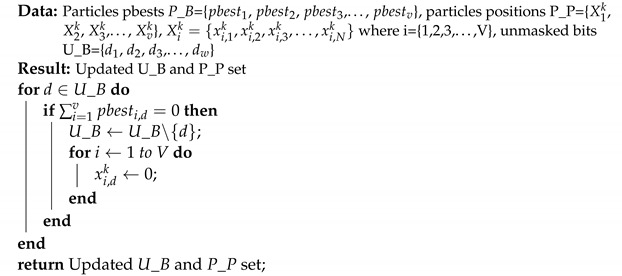


In this algorithm, the pbests of particles are represented by a set, P_B, and the mask is denoted by U_B, given that each element in this set corresponds to an unmasked bit. The U_B set is updated by obtaining information from the pbest of each swarm. A bit is removed from U_B if it is not selected by all pbests in the swarm. During the algorithm’s evolutionary phase, the set is updated. Some bits in U_B are masked each time the mask- update mechanism is run. The mask-update approach ensures a reduced search space, because only the bits in U_B can evolve. The position-updating mechanism can then be rewritten as in (11) [28].
(11)$$x_{i,d}^{k+1}=\left\{\begin{array}{cc} 1-x_{i,d}^{k} & rand\left(\right)<x_{i,d}^{k+1},d\in U\_B\\ x_{i,d}^{k} & rand\left(\right)\geq x_{i,d}^{k+1},d\in U\_B\\ 0 & d\notin U\_B \end{array}\right.$$

According to the third condition, if d∉U_B, the position of that particle is assigned a value of 0, meaning that it is eliminated from the search space to reduce computational time and resources. This improvement can be seen in Figure 2, where the grey blocks show the improved SBPSO strategies.

The overall IDSBPSO-based feature selection procedure can be seen in Algorithm 2. The proposed approach first adopts a search space reduction strategy to reduce the number of features involved in the iteration update, and the mask is updated every μ. *K* iterations, with *K* as the maximum number of iterations.
**Algorithm 2:** Pseudocode of the IDSBPSO-Based Feature Selection Method
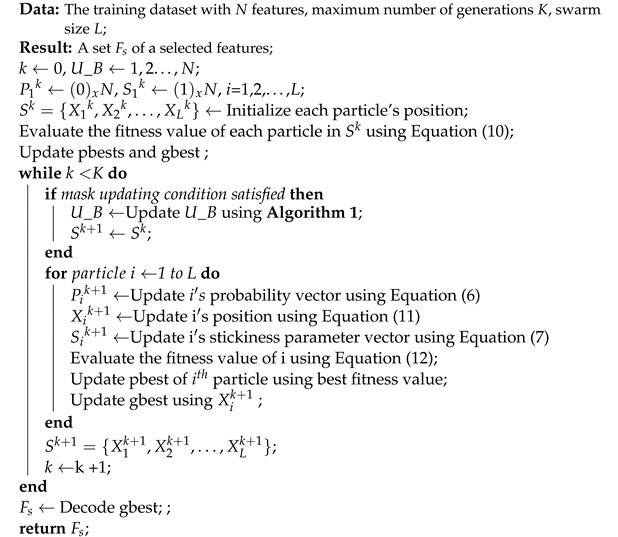


Figure 3 depicts the framework of the proposed model for a network IDS using IDSBPSO. The proposed system comprises a number of phases to obtain good accuracy and network suitability, as explained in this section.

The two IoT datasets included IoTID20 and UNSW-NB15. The IoTID20 dataset was generated in 2020 [29] and contains a total of 83 network features. These network features can be seen in Table 1. There are also three label features in this dataset: binary, category, and sub-category, and four main attacks: Scan, Mirai, denial of service (DoS), and man in the middle (MITM). These attacks and their subcategories can be seen in Table 2.

The UNSW-NB15 is an advanced dataset used for IDS research. It is widely referenced in the literature. The UNSW-NB15 contains 42 network features, as listed in Table 3. There are two label features in this dataset: binary and category. Moreover, nine attacks may be seen in Table 4. Authentication, confidentiality, integrity, and availability are among the security needs targeted by these attacks. Accurate detection of these attacks is critical, as the consequences for IoT applications can be disastrous [30].

Real-life datasets are high-dimensional because they incorporate vibrant information, received from a variety of IoT devices and sensors. When creating an ML model, it is essential to choose a set of meaningful, non-redundant features, because the quality of the features will reduce the performance of the ML classifier [31,32] and the data will be unsuitable for IoT devices to work on. For this purpose, IDSBPSO-based feature optimization was used in this study. The feature optimization problem may be formulated in different ways. In many cases, there is a need to optimise features to reduce computational cost, while also increasing performance accuracy to enhance the generalisation capability. When choosing the best optimization technique based solely on prediction accuracy, performance will vary between the training and test sets [33]. Therefore, there are two main goals in feature selection: to improve classification performance and reduce the number of selected features. In [34], the aggregate fitness function is used to select best features with no change in accuracy, which can be shown in (12).
(12)$$l=a\times l_{1}+\left(1-a\right)\times l_{2}$$
where $l_{1}$ is the error rate, *a* is a constant giving weightage to the terms, and $l_{2}$ is the percentage of selected features, which can be seen in (13).
(13)$$l_{2}=p/n$$
where *p* represents the total number of selected features out of a total of N features. The value of *a* has been adjusted to 0.8 since it was suggested as being between 0.7 and 0.9 [35]. The selected features were then entered into the ML classifier. Random forest classification is used, this being a group of tree-structured classifiers in an ensemble technique. Each tree is built with a decision tree and different bootstrap sample from the original data. Each node of trees only selects a limited selection of features for the split. Out-of-bag (OOB) evaluation, which is an unbiased estimator of generalisation error, is performed on the learning samples that are not selected using the bootstrap. When a new sample needs to be classified after the forest has been built, it is fed into each tree in the forest. Each tree then casts a unit vote for a specific class, indicating the tree’s judgement. When compared to typical ML classifiers, ensemble classifiers are strategies that may be adopted to build a powerful classifier with improved classification accuracy. The mathematical expression representing the model can be seen in (14).
(14)$$C\left(x\right)=sign\sum_{j=1}^{m}\left(C_{j}\left(x\right)\right)$$
where *j* represents each classifier and *m* represents the total number of classifiers included in the classification or voting.

Random forest has the following advantages:It demonstrates excellent performance in accuracy on structured data.It is computationally efficient and can run on large-scale datasets with high dimensions.In most cases, it does not overfit and is robust against noise.It can handle unbalanced datasets.

## 4. Implementation and Evaluation of Results

This section discusses the experimental setup, evaluation metrics used to check the proposed model’s performance, parameter setup, and experimental results, ending with an evaluation of the results of the proposed model.

(a)
**Experimental Setup**
The suggested model’s performance was evaluated on a Dell computer, running Microsoft Windows 10 Professional with Intel (R) Core (TM) i7-6500U and CPU at 2.50GHz 2.60 GHz, 2 cores and 4 logical processors, and 16 GB RAM. Feature selection and classification algorithms were implemented in the Python programming language (version 3.8). Anaconda Navigator was installed on the above-mentioned machine for the experimental setup.(b)
**Evaluation Metrics**
The performance of the proposed ML model may be evaluated using the following parameters: accuracy (AC), precision (PR), recall (RC), and F1-score (F1S) [36]. The F1S is the harmonic mean of PR and RC. Meanwhile, AC, PR, RC, and F1S are computed as follows:
(15)$$AC=\frac{TP+TN}{TP+TN+FP+FN}$$
(16)$$PR=\frac{TP}{TP+FP}$$
(17)$$RC=\frac{TP}{TP+FN}$$
(18)$$F1S=\frac{2\times (PR\times RC)}{PR+RC}$$
where each element in the above equations can be defined as follows:True Positive (*TP*): indicating that both the actual and predicted values are positive.True Negative (*TN*): indicating that both the actual and predicted values are negative.False Positive (*FP*): indicating that the actual value is negative, but the model predicted positive.False Negative (*FN*): indicating that the actual value is positive, but the model predicted negative.In addition, computational time was used as an evaluation parameter to verify the efficiency of the proposed model, as the model is being proposed for energy constrained IoT devices.(c)
**Parameter Setup**
In IDSBPSO, the swarm size (total no. of particles to select best solution) was set at *L* = 20, the maximum number of generations was set at *K* = 50, and the step parameter (determines the stickness ability) was set at *M* = 50, as used in [37]. The parameter for updating mask μ was 0.25, given that this has been found to produce good results. Moreover, inertial weight $n_{s}$ decreases from 0.9 to 0.2 and is calculated using (8). The acceleration constant np increases from 0.5 to 2.5 and decreases from 2.5 to 0.5 for $n_{g}$, using (9) and (10).(d)
**Experimental Results**
The experiment was carried out on the IoTID20 and UNSW-NB15 dataset, using the train test split validation method to conduct a detailed performance evaluation of the ML algorithms. The IoTID20 dataset contained 625,783 instances and the UNSW-NB15 dataset contained 2,540,044. Out of this data, 70% was used for training and 30% for validating the model. The binary classification of performance based on AC, PR, RC, and F1S for both datasets can be seen in Table 5 for the normal and malicious network traffic using the proposed method. From the table, it is clear that for both network traffic datasets, the malicious behaviour is detected with almost 100% accuracy over the 20 runs.Figure 4 illustrates the confusion matrix for the binary classification performance for both the IoTID20 and UNSW-NB15 datasets. From Figure 4, it is clear that FN and FP rates are very low which indicates good accuracy and low false alarm rate. While Figure 5 shows that the particle converges to optimal features rapidly with updated number of iterations using proposed IDSBPSO for both datasets.Meanwhile, Table 6 shows the category classification performance of the proposed model, using the evaluation parameters: AC, PR, RC, and F1S. From the table, it is clear that for both network traffic datasets, mostly attacks are detected with good accuracy except Mirai Ack flooding, Analysis, Backdoor, DoS, and Worms.In addition, Figure 6 and Figure 7 show the confusion matrix for the category classification performance for both the IoTID20 and UNSW-NB15 datasets, respectively. From the figures, it is clear that in IoTID20 dataset, attacks are detected with good accuracy except Mirai Ack flooding, While in UNSW-NB15 attacks such as Analysis, Backdoor, DoS, and Worms are detected with low accuracy of classification. Similarly, Figure 8 shows that the particle converges to optimal features rapidly for both the datasets with the updated number of iterations using the proposed IDSBPSO.Figure 9 subsequently shows the number of selected features from the total number of features in both the IoTID20 and UNSW-NB15 datasets. There are total 83 features in IoTID20 datasets out of which only 30 optimal features are selected for training the model. Similarly, in UNSW-NB15, there are total 42 features out of which 15 have been selected. Figure 10 then illustrate the random forest prediction time in the proposed model for both the binary and category classification of the IoTID20 and UNSW-NB15 datasets. As UNSW-NB15 dataset is larger as compared to IoTID20 dataset, therefore model takes more prediction time on it.(e)
**Evaluation of Results**
Table 7 and Table 8 show the results of comparing the IDSBPSO with PSO-based benchmark methods. These PSO-based benchmark methods include SBPSO [37], DSBPSO [27], Up BPSO (UBPSO) [38], Quantum BPSO (QBPSO) [39], Sequential Forward Selection (SFS) [40] and Sequential Backward Selection (SBS) [40]. The results represent the mean value of 20 runs. The results in bold indicate improved computational time. As it can be seen in the table that IDSBPSO takes less time for FS as compared to other state of the art PSO based methods with almost similar accuracies and number of selected features for both datasets. Also the results indicate the improvement in computational time of IDSBPSO for both datasets.

In the tables, it can be seen that the proposed IDSBPSO performs better in terms of accuracy rate and computational cost compared to most of the other PSO-based feature selection methods. IDSBPSO shows slightly more computational cost compared to SFS and SBS but has higher accuracy. This denoted that IDSBPSO is less efficient in terms of computational cost compared to SFS and SBS but better in terms of accurate prediction compared to SFS and SBS. The proposed model obtains a slightly lower accuracy compared with SBPSO and DSBPSO. This means that in some instances, IDSBPSO may remove some informative features, resulting in decreased accuracy compared to SBPSO and DSBPSO. It can be seen from the accuracy results of IoTID20 and UNSW-NB15 datasets that accuracy on UNSW-NB15 is greater compared to IoTID20 dataset. As UNSW-NB15 is a larger dataset compared to IoTID20 dataset so it may be possible that the proposed approach incorrectly mask some main features from already smaller dataset.

In short, the proposed IDSBPSO algorithm obtains higher accuracy while selecting fewer features with reduce computational cost compared with most of the state of the art PSO-based FS methods.

## 5. Conclusions

In this paper, an improved binary PSO algorithm called IDSBPSO is proposed for feature selection in classification. To improve feature selection performance, two mechanisms were adopted for IDSBPSO: a search space reduction strategy and a dynamic strategy to manage the contributions of momentum, pbest, and gbest to the movement of particles, thereby resulting in a balance between exploration and exploitation during the evolutionary process. The proposed method is used to design an anomaly based intrusion detection system for IoT networks due to its less demanding computational cost. Comparison was made on the basis of accuracy, precision, detection rate, F1 score, and computational time. The experimental results for two IoT network datasets demonstrated the effectiveness and efficiency of IDSBPSO. In most cases, IDSBPSO outperformed benchmark PSO-based feature selection methods by obtaining better or similar accuracy with less number of features. In particular, IDSBPSO significantly reduced computational time, compared with benchmark PSO-based feature selection methods, as it is designed for energy-constrained IoT devices.

Although the proposed IDSBPSO algorithm significantly reduced computational time, compared with the benchmark PSO algorithms it was still found to consume a significant amount of computational time, because a wrapper-based technique require extensive computational time. Category classification accuracy of some attacks is not good. The proposed approach works better for large dimensional datasets while it is not more suitable for those datasets with less dimensions as it removes some informative features from them, which results in lower accuracy. Thus, in future, the authors will seek to improve the accuracy of subcategory classification and further reduce computational time. Moreover, IDSBPSO will be used in other applications, as this research was restricted solely to IoT network security. The performance of the proposed algorithm may also be tested using various other classifiers.

## Figures and Tables

**Figure 1 sensors-22-04926-f001:**
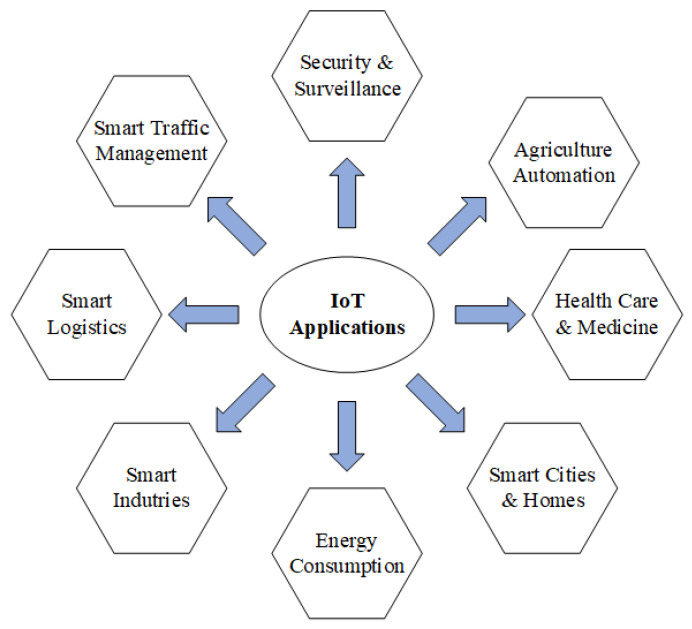
Potential IoT applications.

**Figure 2 sensors-22-04926-f002:**
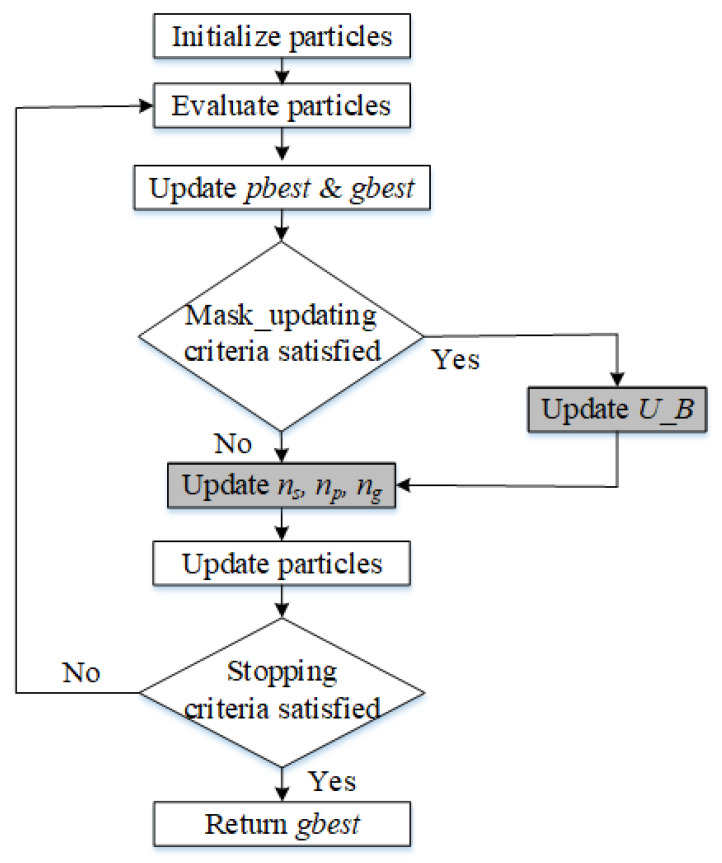
Flowchart for the IDSBPSO.

**Figure 3 sensors-22-04926-f003:**
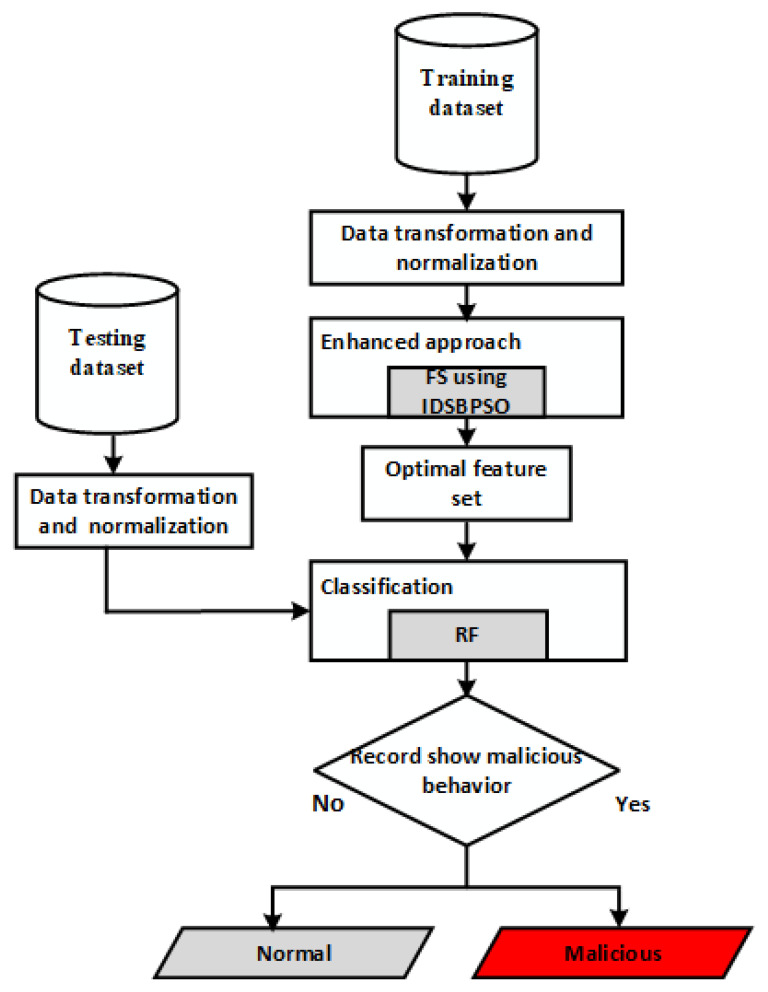
Working architecture of enhanced IDS, using IDSBPSO.

**Figure 4 sensors-22-04926-f004:**
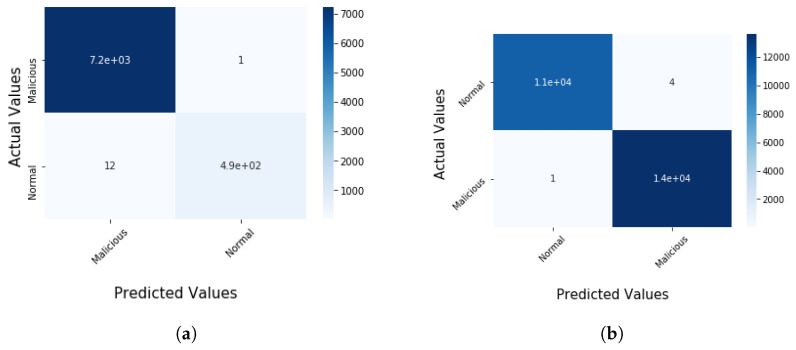
Confusion matrix for binary classification. (**a**) IoTID20 dataset; (**b**) UNSW-NB15 dataset.

**Figure 5 sensors-22-04926-f005:**
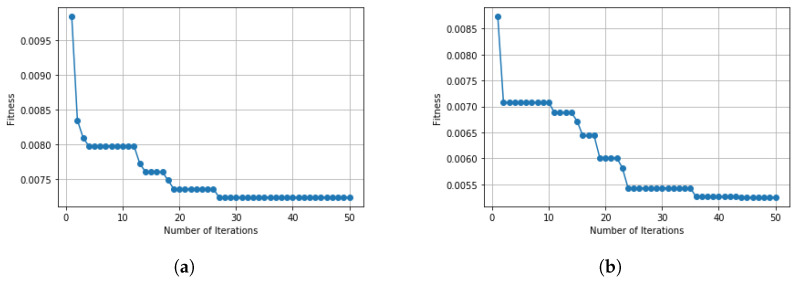
Convergence curve for binary classification. (**a**) IoTID20 dataset; (**b**) UNSW-NB15 dataset.

**Figure 6 sensors-22-04926-f006:**
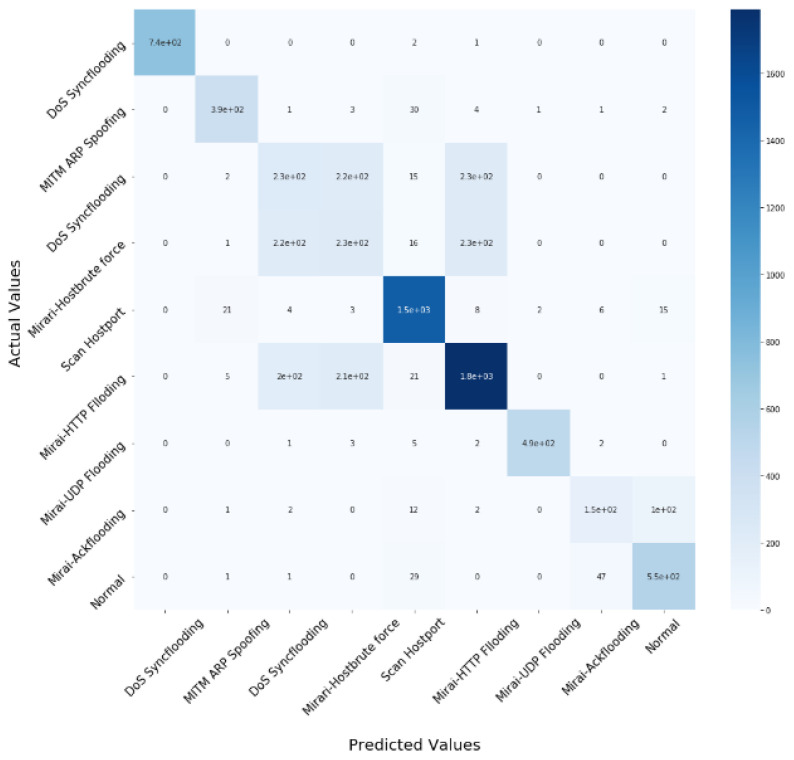
Confusion matrix for the multiclass classification of IoTID20.

**Figure 7 sensors-22-04926-f007:**
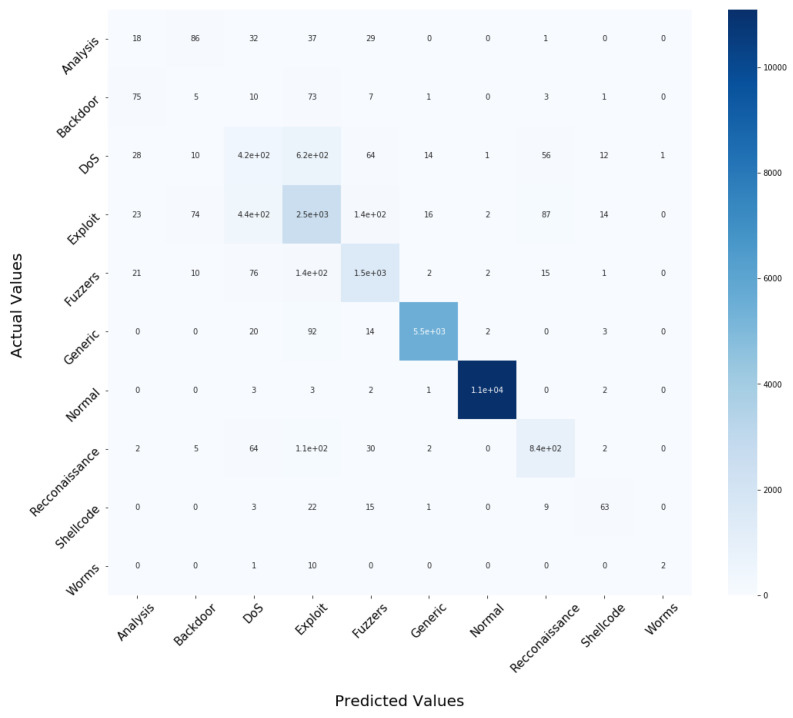
Confusion matrix for the multiclass classification of UNSW-NB15.

**Figure 8 sensors-22-04926-f008:**
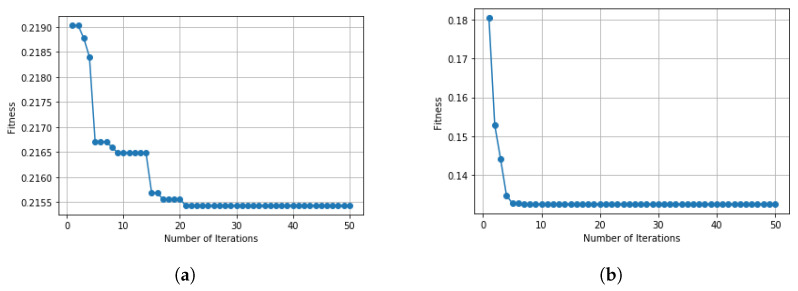
Convergence curve for multiclass classification. (**a**) IoTID20 dataset; (**b**) UNSW-NB15 dataset.

**Figure 9 sensors-22-04926-f009:**
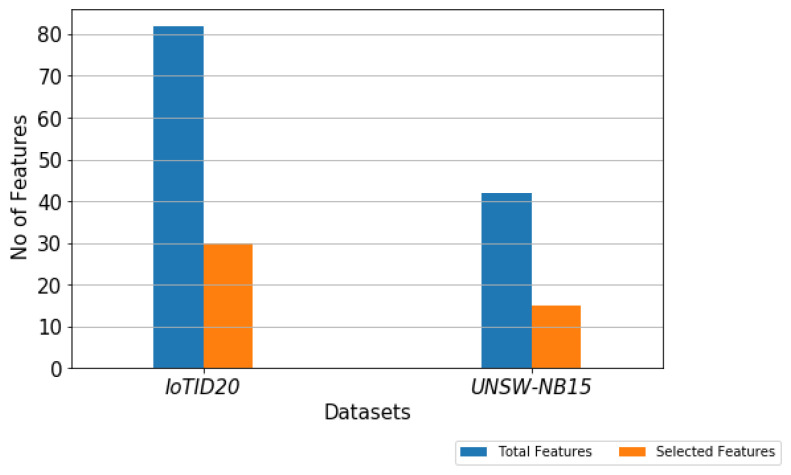
No. of selected features out of total features of IoTID20 and UNSW-NB15.

**Figure 10 sensors-22-04926-f010:**
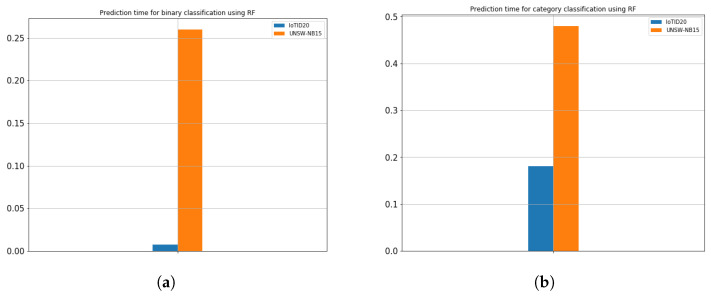
Prediction time (min) for IoTID20 and UNSW-NB15. (**a**) Binary classification; (**b**) Category classification.

**Table 1 sensors-22-04926-t001:** Features of the IoTID20 dataset.

Flow ID	Src IP	Src Port
Dst IP	Dst Port	Protocol
Timestamp	Flow Duration	Tot Fwd Pkts
Tot Bwd Pkts	TotLen Bwd Pkts	TotLen Fwd Pkts
Fwd Pkt Len Min	Fwd Pkt Len Max	Fwd Pkt Len Mean
Fwd Pkt Len Std	Bwd Pkt Len Max	Bwd Pkt Len Min
Bwd Pkt Len Mean	Bwd Pkt Len Std	Active Min
Active Max	Idle Mean	Idle Max
Fwd IAT Tot	Fwd IAT Mean	Fwd IAT Std
Fwd IAT Max	Fwd IAT Min	Bwd IAT Tot
Bwd IAT Mean	Bwd IAT Std	Bwd IAT Max
Bwd IAT Min	Fwd PSH Flags	Bwd PSH Flags
Fwd URG Flags	Bwd URG Flags	Bwd Header Len
Fwd Header Len	Fwd Pkts/s	Bwd Pkts/s
Pkts Len Min	Pkts Len Max	Pkt Len Mean
Pkt Len Std	Pkt Len Var	FIN Flag Cnt
Active Std	SYN Flag Cnt	RST Flag Cnt
PSH Flag Cnt	ACK Flag Cnt	URG Flag Cnt
CWE Flag Count	ECE Flag Cnt	Down/Up Ratio
Pkt Size Avg	Fwd Seg Size Avg	Bwd Seg Size Avg
Fwd Bytes/b Avg	Fwd Pkts/b Avg	Fwd Blk Rate Avg
Bwd Bytes/b Avg	Fwd Pkts/b Avg	Bwd Blk Rate Avg
Subflow Fwd Bytes	Subflow Bwd Bytes	Subflow Fwd Bytes
Subflow Fwd Bytes	Init Fwd Win Bytes	Init Bwd Win Bytes hline
Fwd Act Data Pkts	Fwd Seg Size Min	Active Mean
Idle Std	Idle Max	-

**Table 2 sensors-22-04926-t002:** Attack categories on the IoTID20 dataset.

Scan	Mirai	DoS	MITM
Host Port OS	Brute Force, HTTP Flooding, UDP Flooding	Syn Flooding	ARP Spoofing

**Table 3 sensors-22-04926-t003:** Features of UNSW-NB15 dataset.

dur	proto	service
state	spkts	dpkts
sbytes	dbytes	rate
sttl	dttl	sload
dload	sloss	dloss
sinpkt	dinpkt	sjit
djit	swin	stcpb
dtcpb	dwin	tcprtt
synack	ackdat	smean
dmean	trans_depth	response_body_len
ct_srv_src	ct_state_ttl	ct_dst_ltm
ct_src_dport_ltm	ct_dst_sport_ltm	ct_dst_src_ltm
is_ftp_login	ct_ftp_cmd	ct_flw_htp_mthd
ct_src_ltm	ct_srv_dst	is_sm_ips_ports

**Table 4 sensors-22-04926-t004:** Attack categories of UNSW-NB15.

Generic	Exploits	Fuzzers
DoS	Reconnaissance	Analysis
Backdoor	Shellcode	Worms

**Table 5 sensors-22-04926-t005:** Binary classification of normal and malicious traffic.

Traffic Category	AC	PR	RC	F1S
**IoTID20**				
**Normal**	0.98	1.00	0.98	0.99
**Malicious**	1.00	1.00	1.00	1.00
**UNSW-NB15**				
**Normal**	1.00	1.00	1.00	1.00
**Malicious**	1.00	1.00	1.00	1.00

**Table 6 sensors-22-04926-t006:** Category classification of different attacks.

Traffic Category	AC	PR	RC	F1S
**IoTID20**				
**DoS Sync flooding**	1.00	1.00	1.00	1.00
**MITM ARP Spoofing**	0.92	0.93	0.90	0.92
**Mirai Ack flooding**	0.34	0.35	0.34	0.34
**Mirai-HTTP Flooding**	0.94	0.92	0.96	0.94
**Mirai Host brute force**	0.96	0.95	0.97	0.96
**Mirai-UDP Flooding**	0.80	0.79	0.80	0.80
**Normal**	0.98	0.99	0.97	0.98
**Scan Host port**	0.65	0.73	0.56	0.64
**Scan port OS**	0.85	0.82	0.88	0.85
**UNSW-NB15**				
**Analysis**	0.10	0.11	0.09	0.10
**Backdoor**	0.03	0.03	0.03	0.03
**DoS**	0.38	0.39	0.34	0.37
**Exploits**	0.73	0.70	0.76	0.73
**Fuzzers**	0.84	0.84	0.85	0.84
**Generic**	0.99	0.99	0.98	0.99
**Normal**	1.00	1.00	1.00	1.00
**Reconnaissance**	0.82	0.83	0.80	0.81
**Shellcode**	0.60	0.64	0.56	0.60
**Worms**	0.25	0.67	0.15	0.25

**Table 7 sensors-22-04926-t007:** Results of the evaluation of binary classification.

Method	AC	FS	Computation Time (min)
**IoTID20**			
**SBPSO**	99.80%	30	5.2
**DSBPSO**	99.84%	29	5.2
**UBPSO**	95.20%	34	5.8
**QBPSO**	98.35%	32	5.4
**SFS**	91.00%	25	5.0
**SBS**	86.56%	39	5.1
**IDSBPSO**	99.84%	30	**4.8**
**UNSW-NB15**			
**SBPSO**	99.99%	17	42
**DSBPSO**	99.99%	21	39
**UBPSO**	98.43%	24	35
**QBPSO**	99.90%	18	33
**SFS**	87.64%	14	**30**
**SBS**	85.00%	29	34
**IDSBPSO**	99.95%	13	32

**Table 8 sensors-22-04926-t008:** Results of the evaluation of category classification.

Method	AC	FS	Computation Time (min)
**IoTID20**			
**SBPSO**	79.12%	42	6.4
**DSBPSO**	79.00%	34	6.1
**UBPSO**	78.46%	40	6.4
**QBPSO**	79.03%	38	6.3
**SFS**	62.00%	30	**5.7**
**SBS**	60.89%	45	6.1
**IDSBPSO**	78.46%	37	6.0
**UNSW-NB15**			
**SBPSO**	89.72%	19	45.3
**DSBPSO**	89.57%	19	38.7
**UBPSO**	86.90%	23	30.6
**QBPSO**	89.56%	21	29.9
**SFS**	79.45%	19	**27.0**
**SBS**	75.00%	25	28.2
**IDSBPSO**	89.52%	21	29.6

## Data Availability

The IoTID20 dataset supporting this study was obtained from https://sites.google.com/view/iot-network-intrusion-dataset/home. This is newly developed data, generated in 2020. The UNSW-NB15 dataset was obtained from Kaggle https://www.kaggle.com/datasets/mrwellsdavid/unsw-nb15.

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
