# Peer review of "Enhanced Anomaly Detection System for IoT Based on Improved Dynamic SBPSO"

_sensors, 2022, doi:10.3390/s22134926_

Round 1

Reviewer 1 Report

I see the authors have incorporated the reviewer's comments.
No more comments are from my end. 

Reviewer 2 Report

Dear authors, while the presentation is nice in shape, there are a few comments and/or suggestions to improve the manuscript. Please strongly consider the following suggestions:

  1. The gap in the research should be added after the literature.
  2. The authors must present the limitation of the proposed approach.
  3. The Results and Discussion section should be revised to improve the impact of the paper. The results are abundant but the analysis is not enough. Please reconsider these aspects and consider in the paper text these values.
  4. However, I cannot see deep analysis related to them and cannot understand the meaning of the results. Please add more analysis and extend the discussion section with future research directions.

Minor revisions:

  1. Please add a list with abbreviations and/or nomenclature, for an easily readable paper.

Reviewer 3 Report

The submission describes an improved dynamic sticky binary particle swarm optimization (IDSBPSO) for feature selection, introducing an active search space reduction strategy and several dynamic parameters to enhance the searchability of sticky binary particle swarm optimization (SBPSO). The overall quality of the paper is acceptable. Following are some of the comments which should be addressed in the revised version of this paper:

1. The contribution and innovation of this approach over previous schemes should be clearly stated at the end of the first section.

2. No spaces are required before “Where” after each formula.

3. The author claims that the proposed method uses fewer features. I suggest describing it in detail while introducing the proposed model.

4. The captions of the figures should explain in detail the content of the image.

5. If there are limitations in the proposed approach, I recommend discussing the limits of the approach in the Discussion section.

Reviewer 4 Report

The authors used already used methods, repeating the same solutions. I think those kinds of applications  have no any value, and present nothing to the scientific community. 

There are no real efforts, contribution, or novelty, just an application of PSO algorithm, where in fact deep learning methods can perform better without the burden computational cost of the SI algorithms. 

Round 2

Reviewer 2 Report

No more comments!

Reviewer 4 Report

In fact, I don’t think the contribution and the objective of this paper are enough for publication. Thus, it is hard to accept the paper. The editor may ignore my suggestion and do what you think is good. For me it is very limited contribution with an existing method, with nothing new presented.

This manuscript is a resubmission of an earlier submission. The following is a list of the peer review reports and author responses from that submission.

Round 1

Reviewer 1 Report

The authors propose anomaly detection system for IoT based network using sticky binary particle swarm optimization. I have following comments on the paper:

  1. The motivation behind the paper is not clear. Neither the intent behind using particle swarm optimization method is clear.
  2. There are various deep learning models present in literature which are being used for anomaly detection. The authors should show experimental evaluation with these models.
  3. The related work section is weak. The authors should explain how there model is different then the work already present in this domain.
  4. There are several typographical and grammatical errors in the paper that should be fixed.

Reviewer 2 Report

This paper proposes an improved dynamic sticky binary particle swarm optimization algorithm. Through this approach, an IDS has been designed for detecting malicious data traffic in IoT networks.  When compared to traditional PSO-based feature selection approaches, improved dynamic sticky binary particle swarm optimization algorithm significantly reduces prediction time and computational cost. While the idea is interesting, there are several concerns about this paper.

  1. In the second paragraph of the introduction, the authors mentioned that “Many methods have recently emerged using statistical analysis and ML but designing efficient IDS for IoT devices remains challenging due to the following reasons.” There is a logical error in this sentence. It is suggested to introduce the advantages of statistical analysis and ML.
  2. In the third paragraph of the introduction, the authors mentioned that “Therefore, there are three main approaches to feature optimization. The filter-based approach evaluates features based on some predefined matrics often using information theory. Wrapper and embedded approach evaluate features using ML algorithm.” There are three main approaches, so it is suggested to introduce the reasons for using wrapper-based feature optimization technique.
  3. In related work, there is a problem with the paragraph format. Please indent the first line of each paragraph.
  4. In related work, the defects of the existing literature must be explained. Where is the gap filled by the methods studied in this paper compared with the existing articles?
  5. In the proposed model, the authors most clarify whether the formulas are quoted from existing literatures, or they are proposed by the author. For the previous case, references must be included; for the latter case, the author must clarify e.g., we propose, etc.
  6. In implementation and evaluation of results, there are a large number of figures. The main information expressed in each figure must be explained in detail to facilitate understanding.
  7. In the current version, the latest reference is up to 2021.The authors must refer to latest works in In particular, the following article dealing with high dimensional optimization problems, “Data-driven dynamical control for bottom-up energy Internet system,” in IEEE Transactions on Sustainable Energy, shall be cited, in which the Curriculum Learning method should be discussed in comparison with the SBPSO.

Reviewer 3 Report

Dear Authors. Thanks for submitting your paper to Sensors, an MDPI open-access journal with an impact factor of 3.576.

The paper you submitted concerns the proposal of IDSBPSO for feature selection. I believe that the paper may constitute a contribution to the IoT literature, but it will hardly be accessible to a broad audience at the present state. The paper requires thorough revision and editing since many sentences are redundant or unclear. Also, the results need a broader discussion that is apparently missing in the article.

Please accept the following suggestions to improve the manuscript.

ABSTRACT

Line 1: Delete “for various tasks”. The sentence should already be understandable.

There are many reiterations of the word “various” throughout the abstract and the main body text. Please substitute this word with less generic terms. For example, the sentences would still keep their intended meaning if the word “various” is removed from the abstract.

From line 7 to 12. The sentence is too long and unclear. It should clarify the scope of this study, but it does not. Please rephrase the whole sentence explaining the goal of the study in a more straightforward way that can be accessible also to a non-specialized audience. Especially when line 10 says “and some other dynamic……” it really sounds unclear and non-academic.

Line 13: “In the majority of cases”….again, this sounds vague and non-academic. What percentage of which tests? What specific test? Please be extremely specific and less ambiguous in the description.

INTRODUCTION

Line 18: The first sentence must be rewritten since it does not make much sense. It says that there are higher requests for internet services with the internet. It is like saying that with the invention of cars, we have fuel stations, which is true but also trivial. Can’t you find a more appealing hooking sentence? I suggest you start the paper by citing some recent event, maybe a security breach, that makes this research look more valuable and useful.

Line 30: The authors list security measures that I believe can be shortened by putting three or the main three and then “and so on”.

Line 50 to 62: There is a list of reasons why developing efficient IDS for IoT is challenging. Consider putting the subject in Bold and then a colon (:), because now it is not very clear the subject and the description. For example, the first reason can be written as:

Cyber Security Datasets: The majority of existing……

From lines 73 to 87, the major goals of the paper should be described. I must be honest to say that I could only understand the fourth of four goals. The first three are unclear, and they do not even seem to be goals but methodologies for doing something that is not explained. I suggest the authors rewrite the four goals as objectives of the paper and make sure that their meaning can be easily understandable.

RELATED WORK

This section is quite complete in the sense that it appears that sufficient literature is cited and is related to the field of study. However, it lacks a clear explanation of how this paper position itself among the existing literature. The author should highlight the paper that inspired this one for methodology, field, or dataset and maybe add if they replicate some of their approaches. In a discussion not present in this paper, the author should compare the obtained results with those of a related study cited in the literature review.

PROPOSED MODEL

The background and model elaboration from lines 154 to 163 should be slightly expanded. Left as it is, it is not wrong, but it is hard to understand the theoretical steps made by the authors to elaborate on it. Maybe a little more explanation of PSO and its purpose, its limitation, and the motives for which the author's model should be better. It appears that the description is too short and vague at the moment.

Line 263: about the database selection. When were the databases accessed? Were there updated versions? Were there alternatives? This part lacks a bit of explanation.

Line 269. You write that: “The UNSW-NB15 contains 42 attributes”, but in table 3, you write “selected features from UNSW-NB15”. Then it is not clear if the database contains only 42 attributes or if you extracted 42 attributes from the database. If the attributes were extracted among others, which extraction criteria were utilized?

IMPLEMENTATION AND EVALUATION OF RESULTS

The whole results section is a bit skinny. There are many tables and figures, but they are poorly described and discussed. It is true that if I take the data, you provided and carefully observe all the figures and tables, I may figure out the results, but it should not be like that. The tables and figures should be more extensively explained since, at the moment, this part is quite missing. Also, as said before, the study lacks a critical discussion of the results. A conceptual discussion is as important as a robust analysis. It is essential that the authors clearly state the value they attribute to the results and the limitations of their study, and their approach. This section of the paper needs to be definitely re-elaborated.

CONCLUSION

Line 377 to 379, please rephrase the sentence since it is unclear.

OVERALL THOUGHTS

Overall the paper shows potential, but it lacks a critical discussion by the authors recognizing the limitations of their approach. Also, the manuscript would benefit from professional proofreading since many sentences are unclear and not of academic delivery.

Good Luck with your research!

Reviewer 4 Report

This paper proposes an improved dynamic sticky binary particle swarm optimization algorithm for feature selection that incorporates a dynamic search space reduction strategy and some other dynamic parameters to improve the searchability of sticky binary particle swarm optimization. Hereafter, my comments line-by-line:

Line 26] Fix "net-work" to "network". 
Line 35] Fix "net-work" to "network". 
Line 40] Fix "at-tempting" to "attempting". 
Line 46] Fix "re-searchers" to "researchers". 
Line 73] SBSO has not been previously introduced. Please explain it first.
Line 85] PSO has not been previously introduced. Please explain it first.
Line 98] Please define U2R and R2L.
Line 115] Why the NSL-KDD dataset has not been mentioned in the previous section?
Line 180] Please better explain also the second and third components
Line 196] Fix "ad-here" to "adhere". 
Line 214] Please fix k=0.
Algorithm 2] I think you should clarify how pbests a gbests are updated.
Line 273] Please fix "con-sequences" with "consequences"
Line 284] Please clarify the role of the "fitness function". What is its scope?
Line 297] Please fix "for-est" with "forest".
Line 333] Please clarify the meaning of "swarm size" and "step size".

Overall, I appreciated the paper but I would like to see a comparison with other state-of-art techniques at least in terms of accuracy with respect to the feature number.

Round 2

Reviewer 1 Report

The paper lacks experimental results and comparison with deep learning models which are known to show better performance.

Reviewer 2 Report

No more comments.

Reviewer 3 Report

Dear Authors

Thanks for submitting the revised version of your manuscript

I see many changes have been incorporated, which improved the quality of the manuscript.
Please accept some further comments.

Line 13-14 “with fewer features.” What do you mean exactly? Please, be extremely clear in the abstract.

Line 91-107. Research goals still seem methodologies. Is your research goal to propose a new method of investigation? It seems not, but this is what you declare in this paper passage. Either you change the content of these “goals,” or you change the introduction to these. Instead of saying that those are the key goals/contributions, couldn’t you say the key “methodology steps,” for example? Please find a solution to avoid confusion for the reader.

Table 1. is named: “features of the…..” then the table is labeled, “attributes of…..”. What are these features or attributes?

In conclusion, a small hint on the application in the real world for your model could be introduced to understand your study's contribution better.

Is the limitation of your model the high consumption of computation time? Are there any other limitations in the model? Please be over-critical; the reader will appreciate it.

Lastly, I believe that the manuscript will benefit from proofreading to improve its readability and academic tone, given the importance of the journal.

Good luck with your research. 

Reviewer 4 Report

Your cover letter does not provide clear reference to where the modifications have been made. I can understand it for the typo fixing but not for the other issues. Please update it accordingly.